

# Environmental controls on seasonal ecosystem evapotranspiration/potential evapotranspiration ratio as determined by the global eddy flux measurements

Chunwei Liu[1], Ge Sun[2*], Steve G. McNulty[2], Asko Noormets[3], and Yuan Fang[3]

1. Jiangsu Provincial Key Laboratory of Agricultural Meteorology, College of Applied Meteorology, Nanjing University of Information Science and Technology, Nanjing 210044, China;

2. Eastern Forest Environmental Threat Assessment Center, Southern Research Station, USDA Forest Service, Raleigh, NC 27606, USA;

3. Department of Forestry and Environmental Resources, North Carolina State University, Raleigh,
10
NC 27695, USA.

*Corresponding author:* Ge Sun, 920 Main Campus Dr., Venture II, Suite 300, Raleigh, NC 27606, USA.
gesun@fs.fed.us; (919)5159498 (Phone); (919)5132978(Fax)





**Abstract:** The evapotranspiration/potential evapotranspiration (AET/PET) ratio is traditionally termed as crop coefficient ($Kc$) and has been gradually used as ecosystem evaporative stress index. In the current hydrology literature, $Kc$ has been widely used to as a parameter to estimate crop water demand by water managers, but has not been well examined for other type of ecosystems such as forests and other perennial vegetation. Understanding the seasonal dynamics of this variable for all ecosystems is important to project the ecohydrologcial responses to climate change and accurately quantify water use (AET) at watershed to global scales. This study aimed at deriving $Kc$ for multiple vegetation cover types and understanding its environmental controls by analyzing the accumulated global eddy flux (FLUXNET) data. We examined monthly AET/PET data for 7 vegetation covers including Open shrubland (OS), Cropland (CRO), Grassland (GRA), Deciduous broad leaf forest (DB), Evergreen needle leaf forest (ENF) and Evergreen broad leaf forest (EBF), and Mixed forest (MF) across 81 sites. We found that, except for evergreen forests (EBF and ENF), $Kc$ values had large seasonal variation across all land covers. The spatial variability of $Kc$ was best explained by latitude suggesting site factors has a major control on $Kc$. Seasonally, $Kc$ increased significantly with precipitation in the summer months. Moreover, Leaf Area Index (LAI) significantly influenced monthly $Kc$ in all land covers except EBF. During the peak growing season, forests had the highest $Kc$ values while Croplands (CRO) had the lowest. We developed a series of multi-variatelinear monthly regression models for a large spatial scale $Kc$ by land cover type and season using LAI, site latitude and monthly precipitation as independent variables. The $Kc$ models are useful for understanding water stress in different ecosystems under climate change and variability and for estimating seasonal ET for large areas with mixed land covers.



**Key words**: crop coefficient, evapotranspiration, eddy covariance, modeling, water stress

## 1. Introduction

Evapotranspiration (ET) is one of the major hydrological processes that link energy, water, and carbon cycles in terrestrial ecosystems (Fang et al., 2015;Sun et al., 2011a;Sun et al., 2011b;Sun et al., 2010). In contrast to potential ET (PET) that depends only on atmospheric

water demand (Lu et al., 2005), actual evapotranspiration (AET) is arguably the most uncertain ecohydrologic variable for quantifying watershed water budgets (Baldocchi and Ryu, 2011;Fang et al., 2015; Hao et al., 2015a) and for understanding the ecological impacts of climate and land use change (Hao et al., 2015b), and climate variability (Hao et al., 2014). In recent years, one of the most important research questions of ecohydrology

focused on how ecosystem dynamics, precipitation, AET, and PET interact in different ecosystems at seasonal and long term scales under a changing environment (Vose et al., 2011).

The ratio of AET to PET is traditionally termed as crop coefficient ($Kc$), and has been widely used to as a parameter to estimate crop water demand by water managers (Allen

and Pereira, 2009;Irmak et al., 2013a).However, this parameter has not been well examined for other ecosystems(Zhang et al., 2012;Zhou et al., 2010). The ratio of AET to PET has also been used as an indicator of regional terrestrial water availability, wetness or drought index, and plant water stress (Anderson et al., 2012;Mu et al., 2012).When the AET/PET ratio is close to 1.0, the soil water meets ecosystem water use demand. The ratio of

AET/PET or water stress level can be drastically different among different ecosystems in different environmental conditions, because AET is mainly controlled by climate (precipitation and PET) (Zhang et al., 2001) and ecosystem species composition and





structure (i.e., leaf area index, rooting depth, stomata conductance) (Sun et al., 2011a). The

seasonal PET values for a particular region are generally stable (Lu et al., 2005; Rao et al.,

2011), and deviation of AET/PET from the norm indicates variability in AET, which

responds to precipitation and water availability when PET is stable (Rao et al., 2011).

However, under a changing climate, the AET/PET patterns can be rather complex since

both AET and PET are affected by air temperature and precipitation (Sun et al., 2015a;Sun

et al., 2015b) and corresponding changes in ecosystem characteristics (e.g., plant species

shift) (Sun et al., 2014;Vose et al., 2011).

In the agricultural water management community, the crop coefficient method remains

a popular one for approximating crop water use, despite recent advances in direct ET

measurement methods (Allen and Pereira, 2009;Allen et al., 1998;Baldocchi et al.,

2001;Fang et al., 2015). The $Kc$ is termed as single crop coefficient (Allen et al., 1998;Allen

et al., 2006;Tabari et al., 2013) which is affected by growing periods, crop species, canopy

conductance, and soil evaporation in the field scale (Allen et al., 1998;Ding et al.,

2015;Shukla et al., 2014b). Moreover, $Kc$ can be influenced by soil characteristics,

vegetative soil cover, height, plant species distribution, and leaf area index in a larger

spatial scale (Anda et al., 2014;Consoli and Vanella, 2014;Descheemaeker et al., 2011).

Although the Food and Agriculture Organization of the United Nations provides various

guidelines for several crops (Allen et al., 1998), local measurements are still required to

estimate $Kc$ to account for local crop varieties and for year-to-year variation in weather

conditions (Pereira et al., 2015).

Although the $Kc$ method has been widely used for estimating AET for crops, it has not

been widely used for natural ecosystems for the purpose of estimating AET due to limited



continuous measurements in these systems. However, as discussed earlier, ecologists and hydrologist have started to use $Kc$ to quantify ecosystem stress levels, and consider $Kc$ as a variable rather than a constant. Past studies found that $Kc$ was influenced by the growing stages and leaf area index for maize (Ding et al., 2015;Kang et al., 2003), winter

wheat(Allen et al., 1998;Kang et al., 2003), watermelon (Shukla et al., 2014b), and fruit trees (Marsal et al., 2014b;Taylor et al., 2015). Variations of mid-season crop coefficients for a mixed riparian vegetation dominated by common reed (*Phragmites australis*) could be predicted by growing degree days in central Nebraska, USA(Irmak et al., 2013a). *Kc* ranged from 0.50 to 0.85 for small, open grown shrubs, and from 0.85 to 0.95 for well-

developed shrubland. The *Kc* values had a close logarithmic relationship with the canopy cover fraction in the highlands of northern Ethiopia (Descheemaeker et al., 2011). Overall, the non-agricultural ecosystems such as forests, grasslands and shrublands are heterogeneous in nature and have high soil water availability. Thus, *Kc* values for natural ecosystems have high variability (Allen and Pereira, 2009;Allen et al., 2011).

Therefore, the goal of this study was to explore how Kc varies among multiple ecosystems with various vegetation types over multiple seasons. Another goal was to determine the key biophysical and environmental factors such as latitude, precipitation, and leaf area index that could be used to estimate *Kc*, and if *Kc* can be modeled with a reasonable accuracy in a larger spatial scale. We examined the *Kc* variations for seven land

cover types by analyzing the FLUXNET eddy flux data (Baldocchi et al., 2001;Fang et al., 2015). Specifically, our objectives were to 1) understand the variation of monthly *Kc* for seven distinct land covers by analyzing the influences of environmental factors (e.g., precipitation, site latitude) on *Kc*; and 2) to develop simple land cover-specific regression



models for estimating *Kc* with key environmental factors as independent variables.

Specifically, we developed quantitative relationships between environmental factors and *Kc* by land cover type  using data from FLUXNET sites for 8 croplands(CRO), 13 deciduous broad leaf forests(DB), 5 evergreen broad leaf forests(EBF), 34 evergreen needle leaf forests (ENF), 9 grasslands (GRA), 10 mixed forests (MF), and 2 open shrublands (OS). In-depth understanding of the biophysical controls on Kc for different

ecosystems is important for accurately estimating AET and anticipating the impacts of climate change on ecosystem water stress and water balances.

## 2. Methods

This synthesis study used the LaThuile eddy flux dataset that was developed by FLUXNET

(http://fluxnet.ornl.gov/; Fig. 1), a global network that measures the exchanges of carbon dioxide, water vapor, and energy between the biosphere and atmosphere (Baldocchi et al., 2001). The FLUXNET data (Baldocchi et al., 2001;Baldocchi and Ryu, 2011) have been widely used to understand the evapotranspiration processes and trend (Fang et al., 2015;Jung et al., 2010), develop AET and ecosystem models (Sun et al., 2011b;Zhang et

al., 2016) and map continental-scale ecosystem productivity (Xiao et al., 2014;Zhang et al., 2016).

We used an existing database that was developed from the eddy flux measurements from 81 sites (Fang et al., 2015). According to the International Geosphere-Biosphere Program (IGBP) land cover classification system, these eddy flux sites represent nine land

cover types: open shrubland (OS), cropland (CRO), grassland (GRA), deciduous broad leaf





forest (DB), evergreen needle leaf forest (ENF) and evergreen broad leaf forest (EBF), and

mixed forest (MF). For each eddy flux tower site (Figure 1), we acquired AET and

associated micro-meteorological data, such as vapor pressure deficit (VPD), precipitation

(P), winds speed (WS), net radiation ($R_n$). Reference evapotranspiration($ET_0$) was

calculated by the FAO Penman–Monteith equation as follows(Allen et al., 1998):

$$ET_0 = \frac{0.408\Delta(R_n - G) + \gamma \frac{900}{T + 273} u_2 (e_s - e_a)}{\Delta + \gamma(1 + 0.34u_2)} \qquad (1)$$

where $R_n$ is net radiation at the cover surface (MJ m$^{-2}$ d$^{-1}$), $G$ is soil heat flux (MJ m$^{-2}$ d$^{-1}$),

$T$ is mean air temperature at 2 m height (°C), $u_2$ is wind speed at 2 m height (m s$^{-1}$), $e_s$ is

saturation vapour pressure (kP$_a$), $e_a$ is actual vapour pressure (kP$_a$), $e_s$–$e_a$ is the saturation

vapour pressure deficit (kP$_a$), $\Delta$ is slope vapour pressure curve (kP$_a$ °C$^{-1}$), and $\gamma$ is the

psychrometric constant (kP$_a$ °C$^{-1}$).

  The crop coefficient ($Kc$) is defined as the ratio of the measured AET and the $ET_0$

calculated by equation (1) varies by month and vegetation types (Equation 2).

$$K_c = \frac{ET}{ET_0} \qquad (2)$$

The LAI time series for each tower site were downloaded from the Oak Ridge National

Laboratory  Distributed  Active  Archive  Center  (http://daac.ornl.gov/cgi-

bin/MODIS/GR_col5_1/mod_viz.html). MODIS LAI was derived from the fraction of

absorbed photosynthetically active radiation (FPAR) that a plant canopy absorbs for

photosynthesis and growth in the 0.4–0.7 nm spectral range. LAI is the biomass equivalent

of FPAR. The MODIS LAI/FPAR algorithm exploits the spectral information of MODIS

surface reflectance at up to seven spectral bands. We extracted monthly LAI data for the

time period from 2000 through 2006 across 77 sites using 8-day GeoTIFF data from the




Moderate Resolution Imaging Spectroradiometer (MODIS) land subsets' 1-km LAI global

fields. We estimated monthly LAI for each flux tower by computing the mean of the 8-day

daily values for each month (Fang et al., 2015).

**3. Results**

*3.1. Seasonal variations and long term means of Kc by land cover*

The average monthly *Kc* based on eddy flux data from 2000 to 2007 increased gradually

from January to July and then decreased (Fig. 2). EBF had the highest mean monthly *Kc*

(1.01±0.17) (mean ± standard error) in August. *Kc* for both EBF and ENF varied less

seasonally than other forest types (Fig. 2). Standard errors for GRA, ENF and OS (0.10-

0.17) were larger than other land cover types (0.03-0.10) for April to August. EBF had

higher *Kc* for all seasons than other land covers with a peak value of 0.91 (±0.13) in the

summer season (Fig. 3). In winter seasons, CRO and OS had the lowest *Kc*, 0.25 (± 0.006)

and 0.22 (± 0.004), respectively.

The mean annual *Kc* was 0.39 (± 0.04), 0.47 (± 0.05), 0.79 (± 0.03), 0.45 (± 0.02),

0.57 (± 0.06), 0.45 (± 0.05), and 0.40 (± 0.04) for CRO, DB, EBF, ENF, GRA, MF, and

OS, respectively. Yearly average AET, $ET_0$ and precipitation were higher in EBF than other

land covers (Fig. 4). The precipitation ranking by land cover type was EBF> DB> MF>

GRA> ENF> CRO> OS. Consequently, OS, MF, GRA and ENF had relatively low AET

(376-425 mm). In contrast, CRO had relatively low precipitation with a high $ET_0$.

*3.2. Environmental controls on Kc*

At the annual temporal scale, annual *Kc* was negatively ($p<0.05$) correlated with the

latitude of the sites (Fig.5) for CRO, DB, ENF, GRA and MF with a determination



coefficient ($R^2$) of 0.83, 0.59 and 0.21, 0.72 and 0.52, respectively. For other sites, annual

mean $Kc$ also decreased with the increase in site latitude. Most of the study site latitudes

fell between $30^0$N to $60^0$N except some EBF sites.

At the seasonal scale, the linear relationships between monthly $Kc$ and total monthly

precipitation differed among different land cover types (Fig. 6). Monthly $Kc$ increased with

monthly precipitation in the same ecosystem type with the $R^2$ ranking from high to low:

OS>MF> GRA> ENF>CRO>DB. The monthly $Kc$ for open shrublands (OS) was

especially sensitive to precipitation ($R^2$= 0.69, $p$<0.001). The monthly $Kc$ for EBF was not

as sensitive to precipitation because EBF was generally found in a wet environment with a

peak monthly precipitation of 468 mm. Moreover, $Kc$ for OS, GRA and MF in relatively

drier environments had lower values (Fig. 2). Therefore, $Kc$ was closely related to the

monthly precipitation.

Growing season, site latitude and monthly precipitation affected the monthly $Kc$, in

addition to leaf area index (Fig. 7). $Kc$ was obviously influenced by the leaf area index

(LAI) for all land covers except EBF. The determination coefficients for different land

covers were OS> MF>GRA> ENF>DB>CRO. The LAI could reach 6 $m^2$ $m^{-2}$ in most land

covers, while in OS and CRO the LAI were only 3-4 $m^2$ $m^{-2}$.

*3.3.Kc models*

A series empirical $Kc$ model were developed using a multiple linear regression approach

with precipitation, leaf area index (LAI), and site latitude as independent variables (Table

1).The monthly precipitation, LAI and site latitude influenced $Kc$ ($p$<0.1) for most

ecosystems studied in different seasons except at EBF in summer and fall, and for OS in

the spring. As annual precipitation increases, total leaf area increases, therefore $Kc$ increases for ENF in all seasons and most of the time for DB and MF. As site latitude increases, $Kc$ values were found to decrease in some periods at CRO, DB, EBF and MF

sites. In addition, $Kc$ was closely correlated to LAI, site latitude, and monthly precipitation at ENF in fall and OS in winter with $R^2$ 0.55 and 0.99. All land covers had a peak values 0.53 ($\pm$ 0.04)-1.01 ($\pm$ 0.17) in the summer months. Except for EBF and GRA, $Kc$ values had a close relationship with the monthly precipitation in the summer with $R^2$ ranging from 0.21 to 0.90. The linear relationships were significant for most vegetation types, suggesting

the regression models (Table 1) can be used to estimate monthly $Kc$ if LAI and precipitation are for a specific ecosystem are available.

## 4. Discussion

Our study estimated annual and seasonal crop coefficient ($Kc$) for seven land cover types using measured global eddy flux data. We comprehensively evaluated environmental

controls (i.e., precipitation, LAI, and site latitude) on annual and growing seasons $Kc$ and developed a series of multiple linear regression models that can be used for estimating monthly AET over time and space.

### 4.1. Crop coefficient variation in different seasons

Several recent studies had shown that $Kc$ reached the maximum value in middle of the

growing season in many ecosystems, such as a *P. euphratica* forest in the riparian area (Hou et al., 2010)in a desert environment, a watermelon crop covered with plastic mulch in Florida (Shukla et al., 2014a;Shukla et al., 2014b), soybean in Nebraska (Irmak et al., 2013b), a temperate desert steppe in Inner Mongolia(Zhang et al., 2012). As Fig. 2 shows, most of the land covers had peak $Kc$ during June to August, while the seasonal patterns of

ENF and EBF varied less than other surfaces. Vegetation growth for both the EBF sites is active throughout the year and some EBF sites distributed in the southern hemisphere lead to the stable $Kc$ that varied little. The crop coefficients for early period mid-density fruit trees is about 0.5 (Allen and Pereira, 2009;Allen et al., 1998) which is similar to those found for DB or MF during April and May. In addition, the middle season $Kc$ values for

apple and peach trees with active ground cover were higher than $Kc$ for DB sites during the summer. It is likely that the orchards had higher evapotranspiration rates than natural forests due to irrigation in orchards.

*4.2. Environmental control factors for Kc*

The ecosystem covers and the distributions of the vegetation classes were determined by

the latitude (Potter et al., 1993). Crop coefficient varied predominately by ecosystems, $Kc$ increased as the site latitude decreased for the same land cover (Fig. 5). As the latitude decreased, the temperature and the solar radiation increased and the vegetation characteristics would be different for the same land cover type. Models developed from the FLUXNET data may be best used on flat areas for a given latitude given that eddy

covariance towers were generally installed on flat lands (Baldocchi et al., 2001). For areas with complex topography, the relationship between $Kc$ and site latitude may be more complicated.

Spatial variations of $Kc$ are characteristic of ecosystems, but $Kc$ is also affected by climate factors such as rainfall and temperature. For example, $Kc$ was highly correlated

with precipitation for most land covers (Fig. 6).The rainfall is the major source of soil water and AET in natural ecosystems (Parent and Anctil, 2012). During dry years or periods, a

lack of precipitation may cause a reduction of the leaf area index and *Kc* will decrease to response the ecosystem function. During rainy seasons, as, leaf area index and stomatal conductance of trees and rain-fed crops increases, so does *Kc* (Kar et al., 2006;Zeppel et

al., 2008). Irrigation of cropland is a primary mechanism for increasing yield (Du et al., 2015;Fereres and Soriano, 2007), so the CRO may have a high monthly *Kc* even at sites with a low precipitation. In contrast, *Kc* does not have a close relationship with precipitation under a wet environment. For example, the EBF site had a monthly precipitation as high as 468 mm/month and generally exceeded monthly AET. In an

opposite case for the OS sites, monthly precipitation values were between 0.7 to 69 mm, and *Kc* was highly correlated to monthly precipitation.

Besides precipitation, leaf area index (LAI) also impacted *Kc* in dry and semi-humid area (Kang et al., 2003;Zhang et al., 2012). Unlike precipitation, LAI directly affects *Kc* in AET calculations (Novák, 2012;Tolk and Howell, 2001). Inter-annual *Kc* values are stable

at the GRA and OS sites due to the steady seasonal LAI between years while the plantation forest sites had a more dynamic LAI pattern(Marsal et al., 2014a). As the growth rate of the perennial plants could have large effects on relationship between *Kc* and LAI, long term data are needed to estimate *Kc* as a function of all environmental factors.

*4.3. Modeling the dynamics of Kc*

Our study results are consistent with previous studies that show that the growing stage is a key factor for estimating *Kc* in agricultural crops (Alberto et al., 2014;Allen et al., 1998;Wei et al., 2015;Zhang et al., 2013), fruit trees (Abrisqueta et al., 2013;Marsal et al., 2014b), salt grass (Bawazir et al., 2014) and *Populus euphratica Oliv* forest (Hou et al.,

2010). Additionally, our study showed that $Kc$ fluctuated more dramatically in DB, GRA,

and MF than other land covers in different seasons (Table 1). Studies also show that

monthly leaf resistance that varies over time is important in estimating the seasonal crop

coefficient fora citrus orchard (Taylor et al., 2015). The LAI and total monthly precipitation

varied in both time and space while the site latitude only represents spatial influences on

$Kc$. Thus, the multiple linear regression equations developed from this study take account

of both spatial and temporal changes in land surface characteristics and offer a powerful

tool to estimate of seasonal dynamic $Kc$ for different ecosystems (Table 1).

## 5. Conclusions

To seek a convenient method to calculate monthly AET in large spatial scale, we

comprehensively examined the relations between $Kc$ and environmental factors using eddy

flux data from 81 sites with different land covers. We found that $Kc$ values varied largely

among CRO, DB, EBF, GRA and MF and over seasons. Precipitation determined $Kc$ in the

growing seasons (such as summer), and was chosen as a key variable to calculate $Kc$. We

established multiple linear equations for different land covers and seasons to model the

dynamics of $Kc$ as function of LAI, site latitude and monthly precipitation. These empirical

models could be helpful in calculating monthly AET at the regional scales with readily

available climatic data and vegetation structure information. Our study extended the

applications of the traditional $Kc$ method for estimating crop water use to estimating AET

rates and evaporative stress for natural ecosystems. Future studies should further test the

applicability of the empirical $Kc$ models under extreme climatic conditions.






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




Table 1 Multiple linear regression relationships among crop coefficient and LAI, precipitation and site latitude in different seasons.

| IGBP | season | N | $R^2$ | Kc | b | $a_1$ | $a_2$ | $a_3$ |
|---|---|---|---|---|---|---|---|---|
| CRO | Spring | 24 | 0.16 | 0.31 | 0.242*** | 0.141* | | |
| | Summer | 24 | 0.21 | 0.57 | 0.331** | | | 0.0033* |
| | Fall | 23 | 0.78 | 0.48 | 0.036 | 0.472*** | | |
| | Winter | 21 | 0.36 | 0.26 | 0.920*** | | -0.0141** | |
| DB | Spring | 39 | 0.49 | 0.30 | 0.479** | | -0.0076* | 0.0022*** |
| | Summer | 39 | 0.42 | 0.65 | 0.536*** | | | 0.0011*** |
| | Fall | 39 | 0.13 | 0.60 | 0.462*** | | | 0.0014* |
| | Winter | 39 | 0.15 | 0.30 | 0.713*** | | -0.0094* | |
| EBF | Spring | 15 | 0.25 | 0.74 | 0.875*** | | -0.0050* | |
| | Summer | 15 | - | 0.91 | 0.911*** | | | |
| | Fall | 15 | - | 0.80 | 0.798*** | | | |
| | Winter | 15 | 0.42 | 0.72 | 0.676*** | 0.050* | -0.0050* | |
| ENF | Spring | 96 | 0.39 | 0.37 | 0.225*** | 0.060*** | | 0.0017*** |
| | Summer | 99 | 0.59 | 0.49 | 0.211*** | 0.053*** | | 0.0020*** |
| | Fall | 98 | 0.55 | 0.52 | -0.040 | 0.066*** | 0.0049* | 0.0025*** |
| | Winter | 92 | 0.21 | 0.44 | 0.293*** | 0.084* | | 0.0010* |
| GRA | Spring | 27 | 0.48 | 0.45 | 0.237*** | | | 0.0052*** |
| | Summer | 27 | 0.23 | 0.86 | 0.572*** | 0.110* | | |
| | Fall | 27 | 0.30 | 0.76 | 0.499*** | 0.123** | | |
| | Winter | 27 | 0.26 | 0.41 | 0.256** | | | 0.0038** |
| MF | Spring | 30 | 0.67 | 0.31 | 0.099** | 0.188*** | | 0.0012*** |
| | Summer | 30 | 0.40 | 0.61 | 0.372*** | | | 0.0029*** |
| | Fall | 30 | 0.54 | 0.58 | 0.250*** | 0.071*** | | 0.0018*** |
| | Winter | 30 | 0.13 | 0.33 | 0.961** | | -0.0136* | |
| OS | Spring | 6 | - | 0.23 | 0.230*** | | | |
| | Summer | 6 | 0.90 | 0.35 | -5.419* | | 0.1005* | 0.0026* |
| | Fall | 6 | 0.88 | 0.42 | -9.921* | 0.051* | 0.1828* | |
| | Winter | 6 | 0.99 | 0.14 | -4.919* | 0.629* | 0.0882* | 0.0032* |

Note: N is the number of observations used, $R^2$ the determination coefficient, $Kc_{Ave}$ is the average $Kc$ for seasons. $b$ is the intercept of the multiple linear equation, $a_1$ the coefficient of LAI, $a_2$ the coefficient of site latitude (Absolute values), $a_3$ the coefficient of precipitation. IGBP is the International Geosphere-Biosphere Program land cover classification system: cropland (CRO), deciduous broad leaf forest (DB), evergreen broad leaf forest (EBF), evergreen needle leaf forest

(ENF), grassland (GRA), mixed forest (MF), and open shrubland (OS). ***, **, * stand for $p<0.001$,



$p<0.01$, $p<0.1$. Spring is the month of February, March and April; Summer is the month of May,

June and July; Fall is August, September and October; Winter is November, December and January.



**Figure captions**

Fig. 1 Location of eddy flux sites from which climate and evapotranspiration data are collected.

Fig. 2 The variation of $Kc$ for the different IGBP_code.

Fig.3 Average $Kc$ at spring, summer, fall and winter in different vegetation types.

Fig. 4 Annual total precipitation (P), ET and $ET_0$ in different vegetation types

Fig. 5 The average annual $Kc$ variation at different latitude. (a) stand for cropland (CRO), deciduous

broad leaf forest (DB), evergreen broad leaf forest (EBF), and (b) evergreen needle leaf forest

(ENF), grassland (GRA), mixed forest (MF), and open shrubland (OS). The absolute values of the

latitude were used in EBF in the southern hemisphere sites and all the determination coefficient ($R^2$)

listed in the figure were significant ($p<0.05$).

Fig. 6 Relationships between the average monthly $Kc$ and the total monthly precipitation (P, mm)

for different vegetation surfaces. (a)~(g) represent for cropland (CRO), deciduous broad leaf forest

(DB), evergreen broad leaf forest (EBF), evergreen needle leaf forest (ENF), grassland (GRA),

mixed forest (MF), and open shrubland (OS). All the determination coefficient ($R^2$) listed in the

figure were significant ($p<0.001$)

Fig. 7 Relationships between the average monthly $Kc$ and leaf area index for different vegetation

surfaces. (a)~(g) stand for cropland (CRO), deciduous broad leaf forest (DB), evergreen broad leaf

forest (EBF), evergreen needle leaf forest (ENF), grassland (GRA), mixed forest (MF), and open

shrubland (OS). All the determination coefficient ($R^2$) listed in the figure were significant ($p<0.001$)




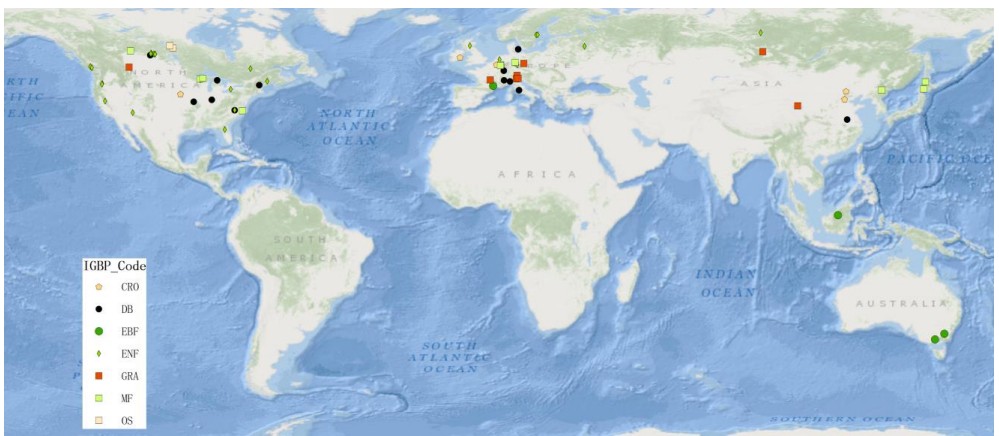

Fig. 1 Location of eddy flux sites from which climate and evapotranspiration data are collected.


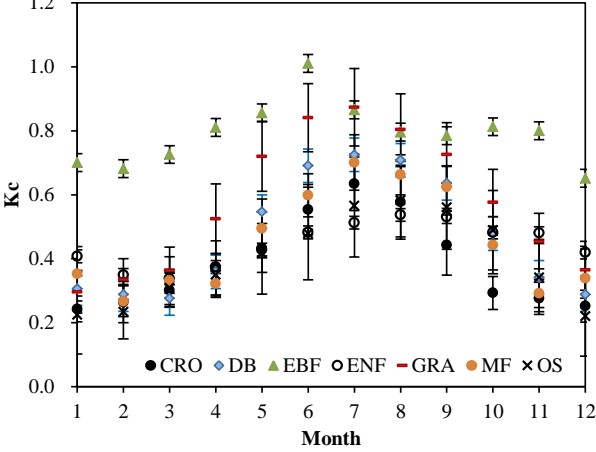

Fig. 2 The variation of *Kc* for the different IGBP_code.





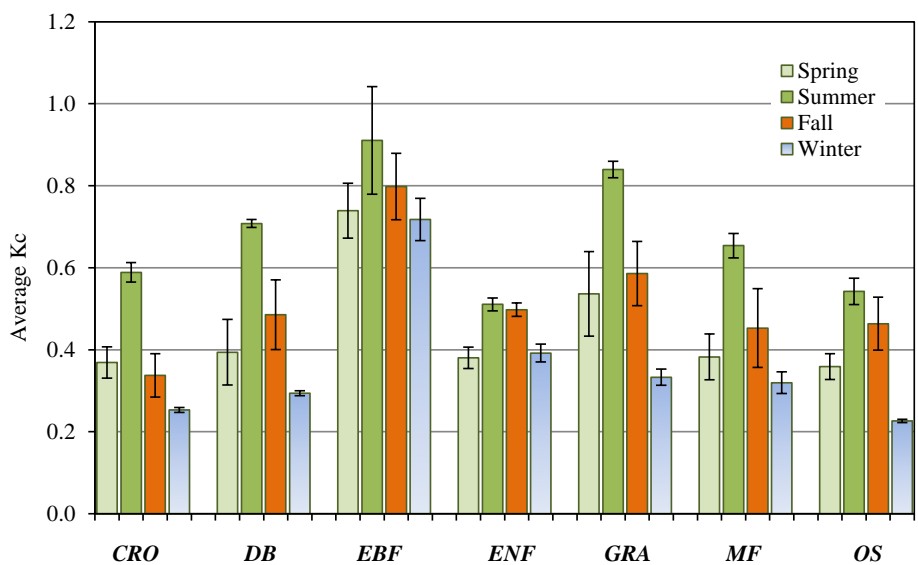

Fig.3 Average *Kc* at spring, summer, fall and winter in different vegetation types.

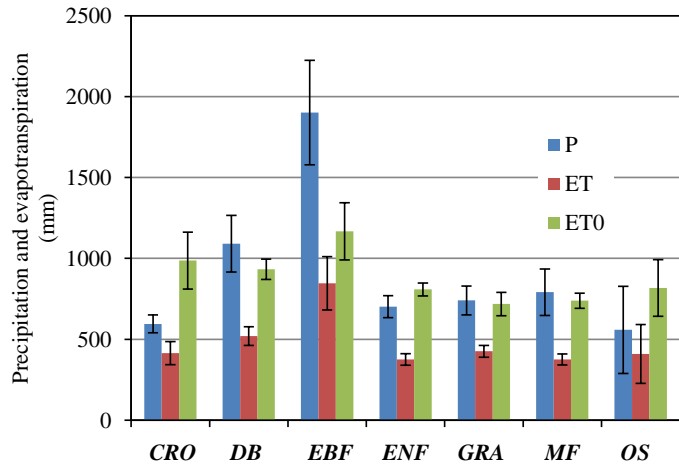

Fig.4 Annual total precipitation (P), ET and $ET_0$ in different vegetation types





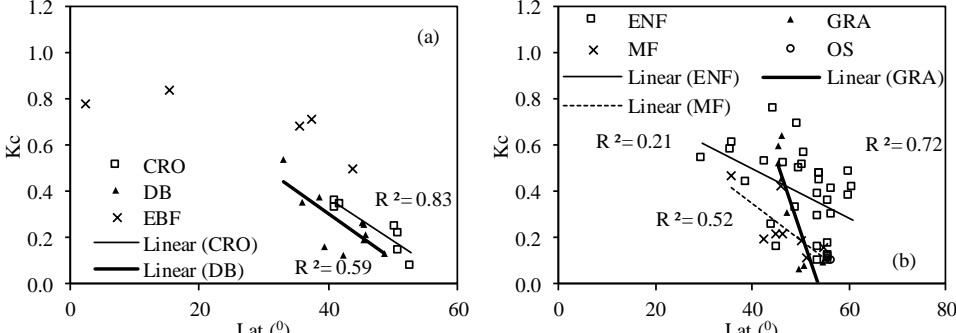

Fig. 5 The average annual *Kc* variation at different latitude. (a) stand for cropland (CRO), deciduous broad leaf forest (DB), evergreen broad leaf forest (EBF), and (b) evergreen needle leaf forest (ENF), grassland (GRA), mixed forest (MF), and open shrubland (OS). The absolute values of the latitude were used in EBF in the southern hemisphere sites and all the determination coefficient ($R^2$) listed in the figure were significant ($p<0.05$).






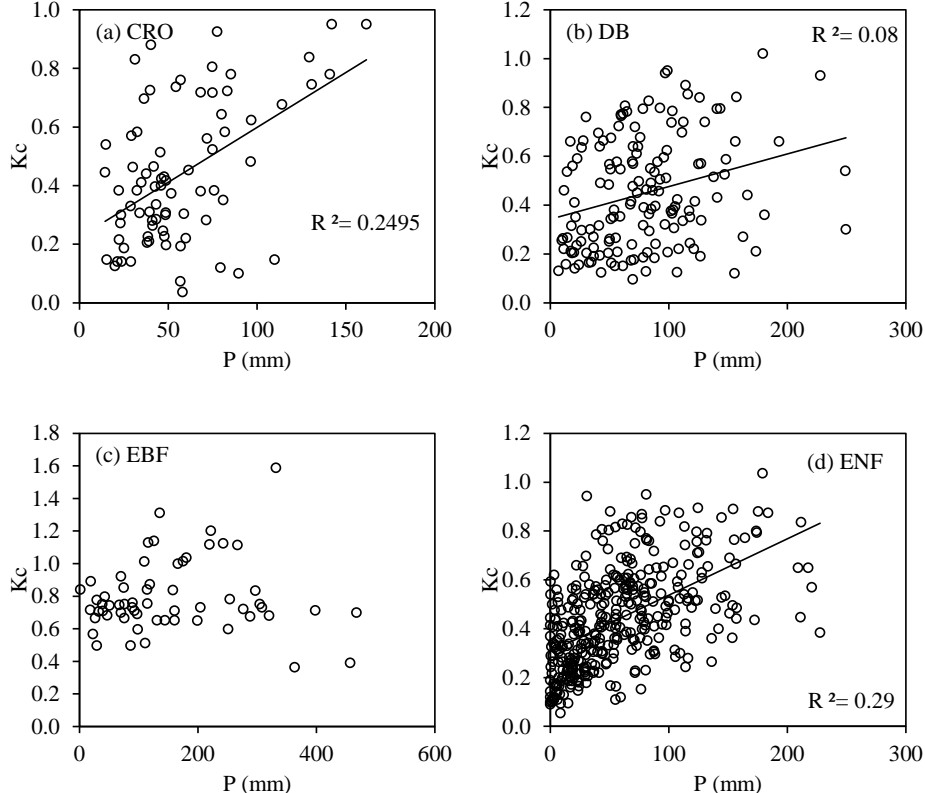






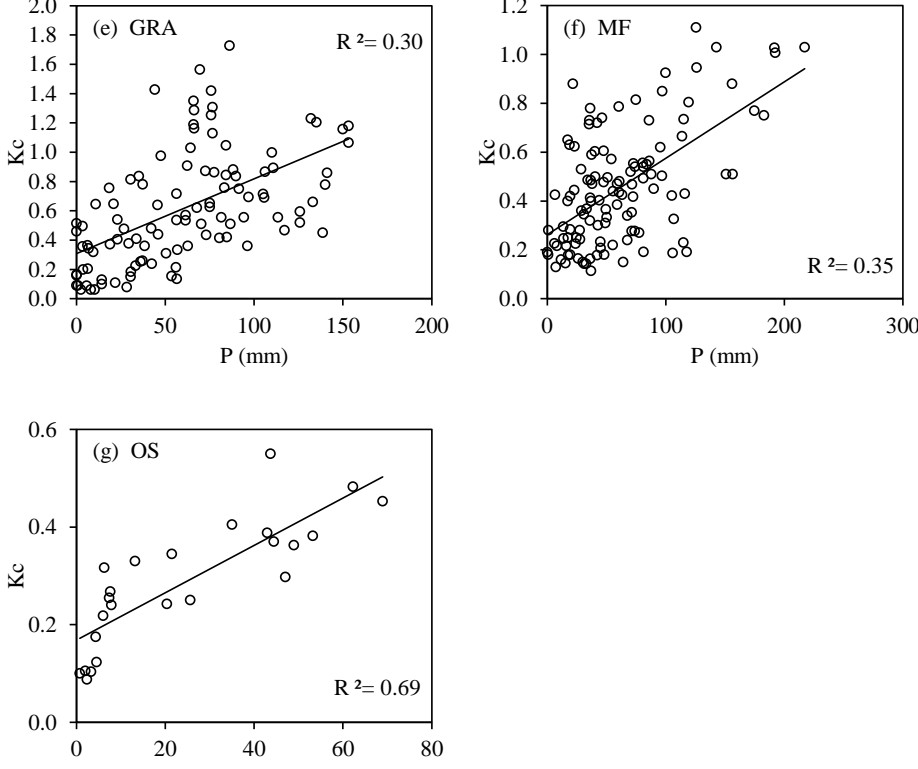

Fig. 6 Relationships between the average monthly *Kc* and the total monthly precipitation (P, mm)

for different vegetation surfaces. (a)~(g) represent for cropland (CRO), deciduous broad leaf forest

(DB), evergreen broad leaf forest (EBF), evergreen needle leaf forest (ENF), grassland (GRA),

mixed forest (MF), and open shrubland (OS). All the determination coefficient ($R^2$) listed in the

figure were significant ($p < 0.001$)





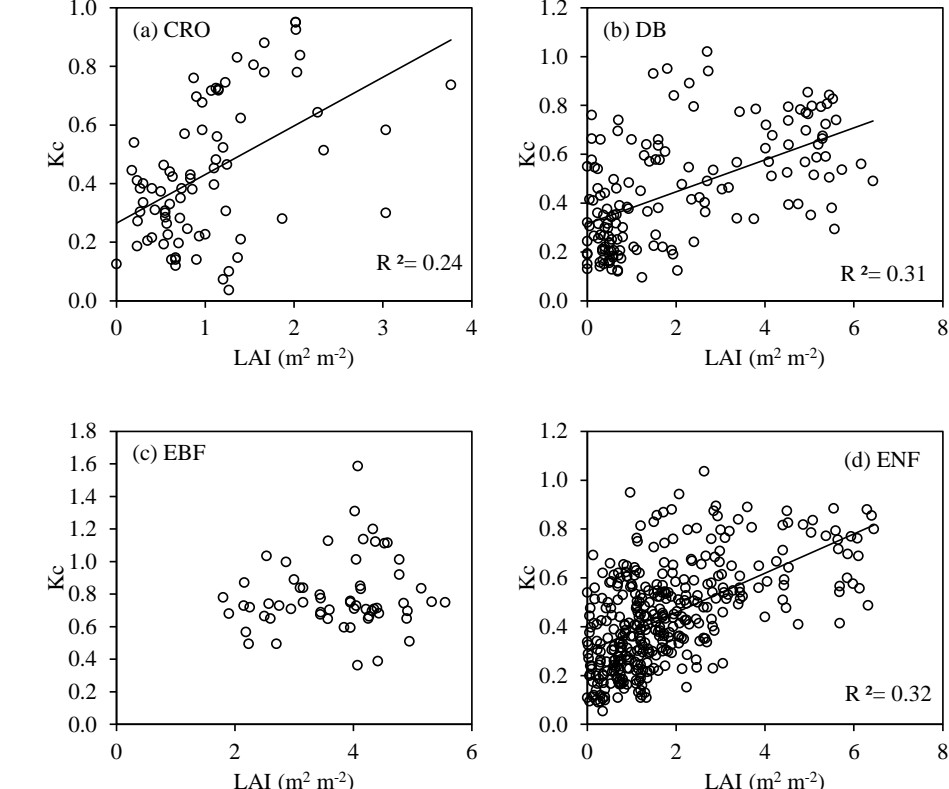





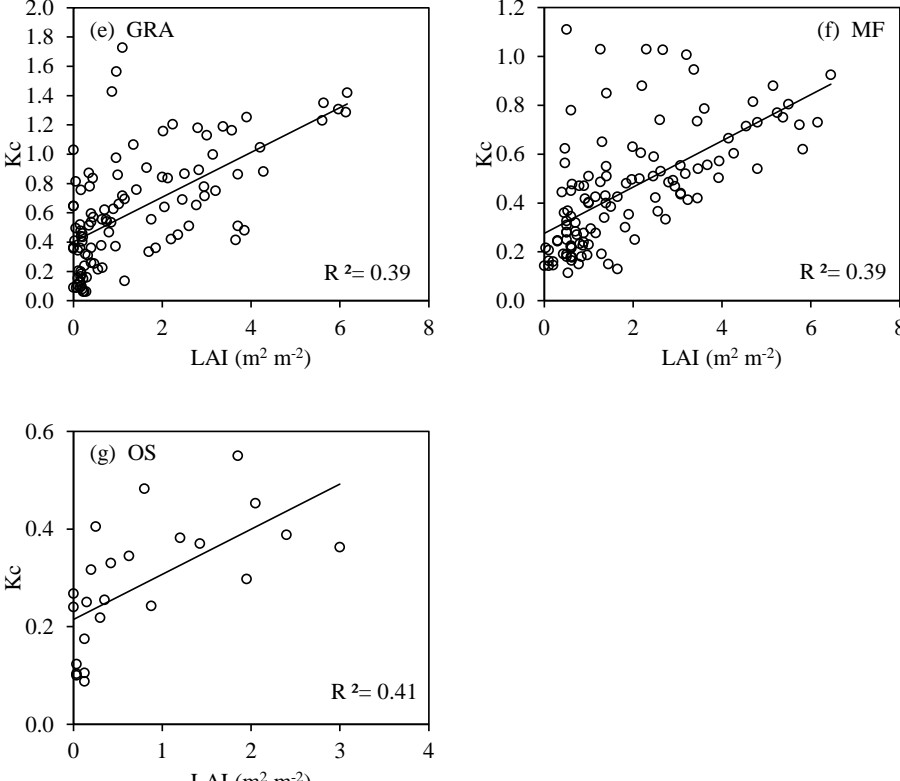

Fig. 7 Relationships between the average monthly *Kc* and leaf area index for different vegetation

surfaces. (a)~(g) stand for cropland (CRO), deciduous broad leaf forest (DB), evergreen broad leaf

forest (EBF), evergreen needle leaf forest (ENF), grassland (GRA), mixed forest (MF), and open

shrubland (OS). All the determination coefficient ($R^2$) listed in the figure were significant ($p<0.001$)
