# Peer review of "Environmental controls on seasonal ecosystem evapotranspiration/potential evapotranspiration ratio as determined by the global eddy flux measurements"

_Hydrology and Earth System Sciences, 2016_

## Referee Comment (RC1) · Anonymous Referee #1 · 18 Jun 2016

The manuscript "Environmental controls on seasonal ecosystem evapotranspiration/potential evapotranspiration ratio as determined by the global eddy flux measurements" by Liu et al. explores the possibility to extend the use of 'crop coefficients' from crops (as proposed by FAO) to natural vegetation. The manuscript also attempts to estimate such coefficients based on eddy covariance data from several locations in the world.

The idea is interesting, as potentially one could estimate the actual evapotranspiration from easy-to-obtain basic meteorological data, geographical location and vegetation

type. Nevertheless, I think the manuscript does not deliver what it promises. The bulk of results focuses on the correlation between crop coefficient and climatological data or basic ecosystem properties (e.g., LAI), presenting mostly expected relations. The impact of this work would be greatly enhanced should the authors really tested their approach, by, e.g., calculating the crop coefficients on the basis of their multivariate linear model and basic ecosystem and climatic data and comparing the results with the estimates from eddy covariance data.

Aside from the specific results, the manuscript and methodology suffer from several, mostly addressable, issues: - Time scales are important, as some processes may be relevant at specific scales. Yet, it remains unclear throughout the manuscript at what time scales the method is applied and to which scales the data refer. Specifically: is the method applied at the annual time scale or at the monthly time scale? Are the data shown monthly (or annual) averages for a specific year or across several years? To what time scales do the following statements refer? L 59 (subdaily to seasonal?), L 69 (decades to centuries?), L 88 (within a certain developmental stage?), L 138 (daily, monthly, annual or multi-annual means?) - Most of the eddy covariance sites are mid-to-high latitude sites, where most likely temperature and solar radiation are the limiting factors for evapotranspiration during part of the year, potentially even leading to leaf shedding in deciduous ecosystems or absence of crops in some cropping systems. Hence, rather than working at the annual scale (as suggested by L 173), it would be probably more meaningful to restrict the analyses to months in which vegetation indeed drives actual evapotranspiration, e.g., on the basis of LAI dynamics or an indicator based on temperature/day length. This would also mean considering dry/wet seasons in the few tropical ecosystems. - More in general, this work would benefit from more attention to the main mechanisms defining actual and potential evapotranspiration. Accounting for seasonality is an example in this sense. Another example is the role of temperature, which appears not relevant in the introduction and method description, yet impacts both potential and actual evapotranspiration in a nonlinear way, directly and indirectly (e.g., via vegetation). Finally, it would be helpful to have some

more information on the crops – if annual summer crops, their winter Kc (L 164) represents other, non-vegetation related, mechanisms. - Finally, the dataset available to the authors is heavily dominated by temperate and boreal ecosystems, with very few tropical sites. I am well aware that only limited eddy covariance data are available from low-latitude sites. Nevertheless, I think that the authors should either limit their attention to temperate and boreal sites (underlining this limitation in their results) or obtain at least few more datasets from the currently under-represented ecosystems/regions. This second approach may require moving beyond FLUXNET data, but may greatly enhance the impact of the work.

Minor issues: - Please use the same symbols and terminology throughout the manuscript (e.g., potential evapotranspiration is later referred to as reference evapotranspiration). - L 63: maximum stomatal conductance may be considered an ecosystem property, but actual stomatal conductance depends not only on vegetation types, but also on soil moisture, VPD, wind speed. - L 149: LAI is not a biomass measure; it is linked to leaf biomass via the specific leaf area, but this parameter varies across ecosystems. - L 159: months are very not meaningful when combining data from northern and southern hemispheres; rather, refer to summer and winter. - L 194: as pointed out on P. 10, LAI and precipitation (and latitude) are not necessarily independent. A justification of the approach is thus necessary. - Whiskers in Figures 2-3 mix different sources of variability – across locations and, for each location, across years. I wonder if it would be more meaningful to distinguish these two aspects.

---

## Referee Comment (RC2) · Anonymous Referee #2 · 23 Jun 2016

General comments

The authors use flux data to calibrate a simple empirical model of the actual to potential evapotranspiration ratio that can be used to calculate AET for other parts of the world, which is a subject appropriate for HESS. I have, however, some doubts about some parts of the methodology and the authors don't really show the potential of the model by applying it. Therefore, I think a major revision is needed.

I get the impression that crop methods (e.g. L 71) are applied to other ecosystems without proper consideration of how the different structure and other properties of those

systems should affect the methods. Eq 2 is constructed for crop and I think it is fine to apply it to other ecosystems for a reference, but it is not correct to use the wind speed at 2 m height measured within closed forest canopies for that calculation. You need to in some way transform the wind data for those sites to open field wind speed or use another parameterization and wind speed at a higher level. As it is done now ET0 is underestimated and Kc is overestimated for the forest sites. For the within land-cover type evaluation it might not make a big difference but in the comparison of Kc levels between ecosystem (e.g. L 166-167) it will matter.

Specific comments

To call latitude an environmental factor (L 102) is questionable though it has a direct connection to the variation of the day length and incoming radiation over the year. Other environmental variables like temperature have some relationship to latitude and it would be better to use those or at least acknowledge that latitude is a proxi for those. This is somewhat done in the discussion (L 231-232), but is should be more clear and stated earlier.

You have some southern hemisphere sites but it seems that you have treated them such as they were expected to have the same monthly variation as the other sites, is that correct?

How were seasonal and yearly Kc calculated, (Sum of ET over months)/(Sum of ET0 over months) or average of monthly Kc values? In my opinion the first method is correct.

In the discussion it would be good to discuss lag effects. There is e.g. a lag between precipitation and soil moisture that can be up to some months. And low soil moisture can lead to a loss of LAI and the low LAI can sustain for a longer period. Precipitation and LAI is included in the environmental variables but not soil moisture but Kc is partly expected to be explained by soil moisture.

[Figure]

It would really have helped the conclusions if the model of KC developed here was applied and verified for AET. The whole conclusion is based around AET estimates but it has not been done. Some year or sites could e.g. been excluded from the calibration and used for validation.

Technical corrections

Be careful to use the same format (italic, subscript) for all the letters in your abbreviations in text, equations, tables and figures. E.g. Kc in L 100, Eq 2 and Fig 2-7.

L 108. Should be "land-cover specific" not "land cover-specific".

L 112. Why do you have an F in all forest abbreviations but not for DB?

L 140. "is slope vapour pressure curve" please write proper English.

L 175. Zeros instead of circular degree symbols are used.

L 187-188. I suggest revising to something like "In addition to growing season, site latitude and monthly precipitation leaf area index affected the monthly Kc" if I understand what you want to say.

L 190-191. I suggest "The LAI range was up to 6 in most land covers, while it only reached 3-4 in OS and CRO".

L 193. "was" not "were".

L 201-202. I would put the numbers within parenthesis in this sentence.

L 206. Delete the first "are".

L 232. "increased" should be replaced by "will in most cases increase" (see specific comment above).

L 267 "for a" not "fora"

Fig 1. Maybe increase symbol size, especially ENF is hard to see.

Fig 2-4. Tell that you are showing mean and standard deviation.

Fig 3. You have not specified what months are included in the different seasons.

Fig 5. Use proper degree sign.

Fig 5 legend. "Variation of annual Kc at ..." might be better.
* * *

---

## Author Comment (AC1) · 30 Jul 2016

*Response to comments by Reviewer #1 on* "Environmental controls on seasonal ecosystem evapotranspiration/potential evapotranspiration ratio as determined by the global eddy flux measurements" *by* Chunwei Liu et al.

***We greatly appreciate the review comments and thank the reviewers for their effort. We have addressed all of the comments and present our response below. The review comments are in regular typeface, while all responses are in italics and boldface.***

The manuscript "Environmental controls on seasonal ecosystem evapotranspiration/ potential evapotranspiration ratio as determined by the global eddy flux measurements" by Liu et al. explores the possibility to extend the use of 'crop coefficients' from crops (as proposed by FAO) to natural vegetation. The manuscript also attempts to estimate such coefficients based on eddy covariance data from several locations in the world.

The idea is interesting, as potentially one could estimate the actual evapotranspiration from easy-to-obtain basic meteorological data, geographical location and vegetation type. Nevertheless, I think the manuscript does not deliver what it promises. The bulk of results focuses on the correlation between crop coefficient and climatological data or basic ecosystem properties (e.g., LAI), presenting mostly expected relations. The impact of this work would be greatly enhanced should the authors really tested their approach, by, e.g., calculating the crop coefficients on the basis of their multivariate linear model and basic ecosystem and climatic data and comparing the results with the estimates from eddy covariance data.

***AUTHOR RESPONSE:   Good suggestion. In the revision, we added new model validation results that examine the multivariate linear model using 30 other sites. The results show that the multiple models can be used for calculating monthly Kc, and monthly AET sufficiently at a large spatial scale and homogeneous ecosystems (Fig.8). (Line 130-133, 216-221, 285-290)***

Aside from the specific results, the manuscript and methodology suffer from several, mostly addressable, issues: - Time scales are important, as some processes may be relevant at specific scales. Yet, it remains unclear throughout the manuscript at what time scales the method is applied and to which scales the data refer. Specifically: is the method applied at the annual time scale or at the monthly time scale? Are the data shown monthly (or annual) averages for a specific year or across several years?

***AUTHOR RESPONSE: The time scale for Kc calculation is really important for model applications. Our method was applied at the monthly time scale. In the establishment of the multiple Kc models, we use monthly average Kc for several years in different sites. We only chose 78 sites to construct the model. Most validation sites have only 1-2 years eddy flux data, which do not represent the whole Kc variations among years.***

To what time scales do the following statements refer? L 59 (subdaily to seasonal?), L 69 (decades to centuries?), L 88 (within a certain developmental stage?), L 138 (daily, monthly, annual or multi-annual means?)

*AUTHOR RESPONSE: The time scale of AET/PET in L60 was annual. We have added this info in the manuscript. Sun et al (2015) focus on the monthly AET/PET (L68), and we made improvement in this study.    The work of Kc mentioned in L88 was for growing seasons and the time scale was mostly daily. The calculation of $ET_0$ were calculated at the daily time scale, and we use a monthly total AET and PET to calculate Kc (L146) in this study.*

Most of the eddy covariance sites are mid-to-high latitude sites, where most likely temperature and solar radiation are the limiting factors for evapotranspiration during part of the year, potentially even leading to leaf shedding in deciduous ecosystems or absence of crops in some cropping systems. Hence, rather than working at the annual scale (as suggested by L 173), it would be probably more meaningful to restrict the analyses to months in which vegetation indeed drives actual evapotranspiration, e.g., on the basis of LAI dynamics or an indicator based on temperature/day length. This would also mean considering dry/wet seasons in the few tropical ecosystems. More in general, this work would benefit from more attention to the main mechanisms defining actual and potential evapotranspiration. Accounting for seasonality is an example in this sense. Another example is the role of temperature, which appears not relevant in the introduction and method description, yet impacts both potential and actual evapotranspiration in a nonlinear way, directly and indirectly (e.g., via vegetation).

*AUTHOR RESPONSE: The manuscript validates against AET using "crop coefficient method" with the eddy flux data in different land covers. However, the ratio of AET/PET is known as Kc in ET simulation for crops, and was influenced by ratio of soil evaporation to ET, canopy resistance, albedo of the canopy surface, and height of the crops in field scale (Allen et al, 1998, FAO 56). Thus, we try to calculate the AET/PET through the analyses on environmental factors including latitude, precipitation, leaf area index in a larger spatial scale. The factors such as temperature and solar radiation were used for PET calculation, and were not independent to AET/PET, as a result, we only chose independent factors to simulate the AET/PET.    (Line 179-183)*

*The monthly AET/PET indeed can be affected by the dry/wet seasons, so we choose precipitation as an independent factor. The latitude is constant in the same site during all months, so we choose average annual AET/PET to analyze the response of AET/PET to latitude, which is a comprehensive factor for different plant communities for the same IGBP land cover type.*

Finally, it would be helpful to have some more information on the crops – if annual summer crops, their winter Kc (L 164) represents other, non-vegetation related, mechanisms.

*Yes,the winter seasons AET for CRO and OS is mainly depended on soil evaporation as the vegetation deforested or harvested. Thus, the different growing seasons for different crops may cause errors in modeling AET.*

Finally, the dataset available to the authors is heavily dominated by temperate and boreal ecosystems, with very few tropical sites. I am well aware that only limited eddy covariance data are available from low-latitude sites. Nevertheless, I think that the authors should either limit their attention to temperate and boreal sites (underlining this limitation in their results) or obtain at least few more

datasets from the currently under-represented ecosystems/regions. This second approach may require moving beyond FLUXNET data, but may greatly enhance the impact of the work.

*Yes,there are still two IGBP land cover types we don't include, which are savannas(SAV) and closed shrubland (CS). The initial dataset contains different number of year, thus, we only chose the monitoring data more than 2 years to establish the multiple AET/PET model. Most of FLUXNET sites are in North America, Europe and Asia, which were mainly in temperate and boreal ecosystems, thus, the more datasets from other experiment sites will be helpful to expand the multiple model. That's a good suggestion for further research.*

Minor issues: - Please use the same symbols and terminology throughout the manuscript (e.g., potential evapotranspiration is later referred to as reference evapotranspiration).

*Yes,we have modified it throughout the manuscript.    (Line 138)*

L 63: maximum stomatal conductance may be considered an ecosystem property, but actual stomatal conductance depends not only on vegetation types, but also on soil moisture, VPD, wind speed.

*Yes,we have modified it. (Line 64)*

L 149: LAI is not a biomass measure; it is linked to leaf biomass via the specific leaf area, but this parameter varies across ecosystems.

*Yes,we have modified it. (Line 155)*

L 159: months are very not meaningful when combining data from northern and southern hemispheres; rather, refer to summer and winter.

*Yes,we have noted it before the submission, however, we forget change Fig.1 since we chose 3 EBF sites in North Hemisphere to modeling Kc. We have modified it in Fig.1 and Line 130.*

L 194: as pointed out on P. 10, LAI and precipitation (and latitude) are not necessarily independent. A justification of the approach is thus necessary.

*Yes,we have improved it. (Line 285-286)*

Whiskers in Figures 2-3 mix different sources of variability – across locations and, for each location, across years. I wonder if it would be more meaningful to distinguish these two aspects.

*Yes,Fig.2 and 3 are from the same data set, and especially Fig.3 is the seasonal Kc for the multiple modeling.*

---

## Author Comment (AC2) · 30 Jul 2016

Response to comments by Reviewer #2 on "Environmental controls on seasonal ecosystem evapotranspiration/potential evapotranspiration ratio as determined by the global eddy flux measurements" by Chunwei Liu et al.

*We appreciate the reviewer' insightful comments. We have addressed all of the comments and present our response below. The review comments are in regular typeface, while all responses are in italics and boldface.*

General comments

The authors use flux data to calibrate a simple empirical model of the actual to potential evapotranspiration ratio that can be used to calculate AET for other parts of the world, which is a subject appropriate for HESS. I have, however, some doubts about some parts of the methodology and the authors don't really show the potential of the model by applying it. Therefore, I think a major revision is needed.

*AUTHOR RESPONSE: We have validated the multivariate linear model using 30 additional study sites. The results showed that the multiple models can be used for monthly Kc calculation, and can be used for monthly AET calculation for large spatial scale and homogeneous ecosystems (Fig.8). (Line 130-133, 216-221, 285-290)*

I get the impression that crop methods (e.g. L 71) are applied to other ecosystems without proper consideration of how the different structure and other properties of those systems should affect the methods. Eq 2 is constructed for crop and I think it is fine to apply it to other ecosystems for a reference, but it is not correct to use the wind speed at 2 m height measured within closed forest canopies for that calculation. You need to in some way transform the wind data for those sites to open field wind speed or use another parameterization and wind speed at a higher level. As it is done now ET0 is underestimated and Kc is overestimated for the forest sites. For the within land-cover type evaluation it might not make a big difference but in the comparison of Kc levels between ecosystem (e.g. L 166-167) it will matter.

*AUTHOR RESPONSE: The PET (ET0) is calculated using the measured wind speed without calibration to 2m height. The reason is that every site has different vegetation types and the height of the wind speed sensors is also different. Thus, we use the measured wind speed to calculate the PET. This treatment indeed is different from reference ET method, especially for forest land cover types. As the reviewer indicated, PET calculated in this study is larger than the reference ET for forests, so the Kc may be underestimated.*

*However, since future applications will be based on the AET/PET response to environment factors, the multi-variant models still could be used for calculating monthly AET in forest ecosystems as long as the PET is measured at the top of the canopy or using wind speed measured at the 2 m height at a standard weather station. In addition, in many application cases, PET is estimated without windspeed in lieu of FAO Reference ET, for example using temperature based method, so it is not an issue in regional applications. We provided more discussion on this issue in Line 142.*

Specific comments

To call latitude an environmental factor (L 102) is questionable though it has a direct connection to

the variation of the day length and incoming radiation over the year. Other environmental variables like temperature have some relationship to latitude and it would be better to use those or at least acknowledge that latitude is a proxy for those. This is somewhat done in the discussion (L 231-232), but is should be more clear and stated earlier.

*AUTHOR RESPONSE: Revised in Line 179-183.*

You have some southern hemisphere sites but it seems that you have treated them such as they were expected to have the same monthly variation as the other sites, is that correct?

*AUTHOR RESPONSE: Thanks for the suggestion. We have corrected it in Fig.1 and Line 130.*

How were seasonal and yearly Kc calculated, (Sum of ET over months)/(Sum of ET0 over months) or average of monthly Kc values? In my opinion the first method is correct.

*AUTHOR RESPONSE: The monthly Kc should be ratio of total monthly ET to total monthly ET0. We revised it in Line 146-150.*

In the discussion it would be good to discuss lag effects. There is e.g. a lag between precipitation and soil moisture that can be up to some months. And low soil moisture can lead to a loss of LAI and the low LAI can sustain for a longer period. Precipitation and LAI is included in the environmental variables but not soil moisture but Kc is partly expected to be explained by soil moisture.

*AUTHOR RESPONSE: Good points and suggestions. The lack of soil moisture do decrease AET, and there will be a time lag after the precipitation occurring. However, we found little improvement in the model using a lagged Preci. Since soil moisture is not measured in many applications we opt not using it. We included some discussion on this (Line 265-268).*

It would really have helped the conclusions if the model of KC developed here was applied and verified for AET. The whole conclusion is based around AET estimates but it has not been done. Some year or sites could e.g. been excluded from the calibration and used for validation.

*AUTHOR RESPONSE: We have validated the multivariate linear model using 30 other sites. The results showed that the multiple models can be used for monthly Kc calculation, and thus can be used for monthly AET calculation for large spatial scale and homogeneous ecosystems (Fig.8). (Line 130-133, 216-221, 285-290)*

Technical corrections

Be careful to use the same format (italic, subscript) for all the letters in your abbreviations in text, equations, tables and figures. E.g. Kc in L 100, Eq 2 and Fig 2-7.

*AUTHOR RESPONSE: We have modified it. (Line 102,117 Eq 2 and Fig 2-7,)*

L 108. Should be "land-cover specific" not "land cover-specific".

*AUTHOR RESPONSE: Corrected (Line 110)*

L 112. Why do you have an F in all forest abbreviations but not for DB?

*AUTHOR RESPONSE: we have corrected it to DBF.*

L 140. "is slope vapour pressure curve" please write proper English.

*AUTHOR RESPONSE: we have corrected it (Line 145)*

L 175. Zeros instead of circular degree symbols are used.

*AUTHOR RESPONSE: we have corrected it (Line 190)*

L 187-188. I suggest revising to something like "In addition to growing season, site latitude and monthly precipitation leaf area index affected the monthly Kc" if I understand what you want to say.

*AUTHOR RESPONSE: we have corrected it (see Line 199)*

L 190-191. I suggest "The LAI range was up to 6 in most land covers, while it only reached 3-4 in OS and CRO".

*AUTHOR RESPONSE: we have corrected it (Line 228)*

L 193. "was" not "were".

*AUTHOR RESPONSE: we have corrected it (Line 231)*

L 201-202. I would put the numbers within parenthesis in this sentence.

*AUTHOR RESPONSE: we have corrected it (Line 240)*

L 206. Delete the first "are".

*AUTHOR RESPONSE: we have corrected it (Line 244)*

L 232. "increased" should be replaced by "will in most cases increase" (see specific comment above).

*AUTHOR RESPONSE: we have corrected it (Line 274)*

L 267 "for a" not "fora"

*AUTHOR RESPONSE: we have corrected it (Line 314)*

Fig 1. Maybe increase symbol size, especially ENF is hard to see.

*AUTHOR RESPONSE: we have corrected it (Fig. 1)*

Fig 2-4. Tell that you are showing mean and standard deviation.

*AUTHOR RESPONSE: we have corrected it (Fig. 2-4)*

Fig 3. You have not specified what months are included in the different seasons.

*AUTHOR RESPONSE: we have corrected it (Fig.3)*

Fig 5. Use proper degree sign.

Fig 5 legend. "Variation of annual Kc at : : :" might be better.

*AUTHOR RESPONSE: we have corrected it (Fig.5)*

---

## Author Response (AR2)

Response to comments by editor and reviewers on "Environmental controls on seasonal ecosystem evapotranspiration/potential evapotranspiration ratio as determined by the global eddy flux measurements" by Chunwei Liu et al.

*We greatly appreciate the review comments and thank the reviewers for their effort. We have addressed all of the comments and present our response below.*

The second round of review came back with widely divergent referee judgments. My own judgment is that the paper is an important one, and that it has improved considerably in this revision. I also believe that in the commented version that is attached I was able to address the language issues. The positive reviewer has, nonetheless, come with a number of constructive suggestions that would further improve the manuscript. The more negative reviewer also provided several constructive comments. I believe that address in all of these issues are relatively straightforward to implement (especially relative to the changes made in the last revision). I am therefore returning the manuscript with an instruction for further revisions noted below. If you feel that you are able to make these changes, I will look at the revised version and make my editorial decision without further referee review.

Comments to address from the reviewer who recommended rejection:

The validation of the proposed approach (included as per both reviewers' suggestions) show that the proposed approach does not provide satisfactory results in 3 out of 7 ecosystems (see Fig. 8).

*AUTHOR RESPONSE: In the revision, we revised dome discussion on the model validation results. The results were less satisfied in CRO, EBF and OS (Line 235, 315-322) and we offered some explanations.*

The results still denote a lack of understanding of the mechanisms driving evapotranspiration in different ecosystems – e.g., the lack of leaves or even plants (for crops) during the winter months. Despite some suggestions on how to handle this issue in the previous round of reviews, the author did not address this point. Rather, in the result section, the authors discuss patterns of average annual Kc, which is not very meaningful when considering temperate and boreal sites (as apparent from Fig.1). Further, they recognize that the seasonal pattern of Kc in evergreen vegetation is more stable in deciduous ecosystems, yet they fail to state why this is the case (needles are retained throughout the year, as opposed to the situation for deciduous trees).

*AUTHOR RESPONSE: We have stressed the phenology of different ecosystem in this revision. We discussed seasonal changes of Kc in this revision in relation to LAI (Line 262-266). A Fig on the monthly AET and PET was added.*

Second, the validation now introduced as part of the revisions, clearly shows that in several ecosystems the approach does not work very well (Fig. 8). The modelled vs.

simulated evapotranspiration rates, while correlated, do not fall on the 1:1 line (or at least near it) in at least in 3 out of the 7 ecosystems, leading to overestimated or underestimated values. Interestingly, one of the ecosystems in which the approach is not working is the crops, where the FAO model was developed.

*AUTHOR RESPONSE: Yes, the results were less satisfactory in CRO, EBF and OS. The under-estimation of CRO modeling was 50 percent lower compared to measured. This error may be because the crops were irrigated during water deficit. The model does not account for added water of irrigation. Meanwhile, the OS has a large proportion of bare soil with low soil water content resulting in an overestimate in modeling ET. The low number sample size (fewer sites than other ecosystem types) may cause a low accuracy of validation in OS and EBF. (Line 315-322)*

Third, there are several unclear or incorrect claims. I report here some examples:
L 64 How can PET be considered stable when (the author acknowledge) it depends on temperature and precipitation?

*AUTHOR RESPONSE: Yes, the seasonal PET values vary by season. We meant to say that PET values are rather stable in the same season among different years. We have clarified the statement. (Line 64-65).*

L 90 The FAO approach has been used for many more crops (and the Kc values are tabulated in Allen 1998, for each and every growth stage).

*AUTHOR RESPONSE: Yes, we have modified it. (Line 95)*

L 129 How were the 'validation sites' selected? Where are they located?

*AUTHOR RESPONSE:   We used 30 sites (not used for model developed) with one or two years of data used for model validation. The sites are distributed in the Northern Hemisphere (Latitude between 29-71, and longitude between -125 - 148) (Line 133)*

Fig. 5: I am confused by the evergreen broadleaf forest at 60 deg N (one of the two sites among the evergreen broadleaf forested ecosystems). Probably it would be worth to provide a bit more information about the sites, particularly when very few (and hence potentially non-representative) are available

*AUTHOR RESPONSE: We tried to presented data from EBF sites located in the Southern Hemisphere. In this revision, we used July data as January if the sites are in the South Hemisphere (Fig 2). Thus, we improved the multiple regression model in Table 1 and Figures 1-7 for EBF.*

Comments to address from the reviewer who recommended publication ( I am not sure

the authors got to see this as the reviewer submitted these confidents in a channel that may not have been available to the authors):

The article of Liu et al. is a needed contribution to studies of evapotranspiration rates across ecosystems and regions. The methodology is consequent and clear. I enjoyed reading the article. The multi-linear models developed by the authors for the ratio of actual evapotranspiration to potential evapotranspiration will be an important tool for hydrologists on the field.

However, some aspects need to be improved. Some key studies are completely missing from their manuscript. Starting by the studies of Budyko (1974) were potential and actual evapotranspiration are put in context. I suggest other important references that could better support the discussion.

*AUTHOR RESPONSE: We have added the reference.    (Line 49)*

I suggest an inclusion of a Figure that compares water and energy use efficiency for all the ecosystems compared by the authors. This comparison would enable a direct comparison of Kc (AET/PET) and evaporative ratio (AET/P) for all the ecosystems evaluated in the manuscript, enriching the discussion. See Van der Velde et al. (2013).

*AUTHOR RESPONSE: The reviewer's suggestion is a good one for future research to understand the control of AET by PET and P at an annual scale. However, this study focuses on Kc – we intended to provide a practical way to estimate ET in a large spatial scale.*

Also, general information on the FLUXNET measurements should be included in the manuscript. See below some typical questions.

Any autocorrelation between latitude, precipitation, LAI if not treated independently? I would say that in these Northern Eurasian latitudes as you move northwards you get more rain?

*AUTHOR RESPONSE: Yes, we have examined the autocorrelations among different variables. Precipitation, latitude and LAI were independent from each other.*

Other aspects along the manuscript are found in detail below:

First paragraph and Line 59-You could mention briefly here that the uncertainty in AET is mainly due to all the factors affecting vegetation AET rates as mentioned by Jaramillo et al., Journal of Hydrology (2013) or Donohue et al. Journal of Hydrology (2007), Hasper et al., Functional Ecology (2015) and by all the climatic and landscape drivers of ET change (See Jaramillo and Destouni, GRL, 2014). Even better, mention the most important.

*AUTHOR RESPONSE: We added these important references in the lines 63, 65 and 71.*

Line 86-89- Shown by Zhang et al. (2001)

*AUTHOR RESPONSE: Yes, we added the reference in line 88.*

Line 122- Upfront, please specify the time period of ET availability from FLUXNET, time-scale, how it was obtained, etc. A brief summary could be useful for the general reader that does not know of eddy-flux ET measurements.

*AUTHOR RESPONSE: We added the time scale in line 140.*

Line 134- in what units are AET and VDP being measured by eddy-flux towers?

*AUTHOR RESPONSE: The unit for AET is LE, MJ $m^{-2}d^{-1}$, and for VPD is 100Pa in the original eddy-flux data. We convert it to mm $d^{-1}$ and kPa.*

Line 139-How did you estimate all these parameters, Rn, slope of sat, etc? What assumptions of Allen et al. 1998 did you apply? Just explain what assumptions did you use, mention the equation numbers in Allen et al. 1998. What height was wind speed measured at in the towers? With what time scale was PET estimated? Was it later aggregated in time to agree with the AET time scale?

*AUTHOR RESPONSE: The Rn is from measured data, the G is calculated as 0.1Rn in daytime and 0.5 Rn at night, the slope of saturation vapour pressure curve is calculated as follows:*

$$\Delta = \frac{4098 \left[ 0.6108 \exp\left( \dfrac{17.27T}{T + 237.3} \right) \right]}{\left( T + 237.3 \right)^{2}}$$

*(Line 142)*

*Most wind speed measured height is above the canopy. The time scale was daily for PET estimation. Yes, we aggregate the daily PET to monthly PET.*

Line 163-Can you send to Supplementary a figure showing how AET and PET vary from month to month. This would enrich the discussion to understand the variations of Kc from month to month!

*AUTHOR RESPONSE: We added the monthly AET and PET in Fig 4 that is helpful to understand the difference at different sites in 12 months.*

Ecosystem Acronyms-Can you mention again in the results what each of these acronyms are, EBF,

GRA, etc, it is difficult to go back to the first explanation every time you use one. Or maybe use

easier-to-understand abbreviations.

*AUTHOR RESPONSE: We have modified it.(Line 174-178)*

Line 174-What do you mean with this?

*AUTHOR RESPONSE: Yes, we have modified it. (Line187)*

Line 217- by ecosystem, Line 220- measurements with

*AUTHOR RESPONSE: Thanks for the tips, we have modified it. (Line 232, 235)*

Line 240-I think this should be stated right from the beginning, that some of the flux measurement sites are in irrigated areas! You know, irrigation has been proven to be driving ET changes at the local and even at the global scale. See Jaramillo and Destouni, Science, 2015.So when plotting the figure I just suggest, irrigation could represent much of the high AET or Kc rates. See Van der Velde et al. (2013). Are there other sites that have irrigation in your study? The irrigation issue should be mentioned in the FLUXNET methods.

*AUTHOR RESPONSE: Yes, we have improved it. (Line 150, 256-259).*

Line 244-You should say that this mainly occurs since as latitude is decreasing, PET increases, but AET increases even more than PET. It is the only way this can happen. This is an interesting finding.

*AUTHOR RESPONSE: Yes, we have improved it. (Line 270-272).*

Line 248- Mention this in the beginning of the FLUXNET methods.

*AUTHOR RESPONSE: Yes, we have improved it. (Line 151).*

Line 260- Sorry to be pushy, but again a more updated study showing the domination role of irrigation on ET such as Jaramillo, F., Destouni, G., 2015.

*AUTHOR RESPONSE: Yes, we have improved it. (Line 150, 256-259)*

Line 266- connect the two sentences

*AUTHOR RESPONSE: Yes, we have improved it. (Line291)*

Line 284- What is leaf resistance?

*AUTHOR RESPONSE: Yes, we have improved it. (Line311)*

Table 1- What do blank spaces mean? Non-significant values?

*AUTHOR RESPONSE: Yes, we have improved it. (Line523)*

Figure 1- Legend is missing

*AUTHOR RESPONSE: Yes, we have improved it. (Figure 1)*

Figure 3- Explain uncertainty bars, are they one std dev?

*AUTHOR RESPONSE: Yes, the bars are standard errors. we have improved it. (Line 537)*

Conclusions- the conclusions should state that the models apply to northern temperate and boreal

latitudes, and that its extrapolation to other tropical and southern latitudes should be explored.

*AUTHOR RESPONSE: Yes, we have improved it. (Line 328).*
Some suggested references that could enrich the literature review and discussion mentioned in this review

-Budyko, 1974. Climate and life. Academic Press.
-Donohue, R.J., Roderick, M.L., McVicar, T.R., 2007. On the importance of including vegetation dynamics in Budyko's hydrological model. Hydrol. Earth Syst. Sci. 11, 983–995.
-Hasper, T.B., Wallin, G., Lamba, S., Hall, M., Jaramillo, F., Laudon, H., Linder, S., Medhurst, J.L., Räntfors, M., Sigurdsson, B.D., Uddling, J., 2015. Water use by Swedish boreal forests in a changing climate. Funct. Ecol. n/a-n/a. doi:10.1111/1365-2435.12546
-Jaramillo, F., Destouni, G., 2015. Local flow regulation and irrigation raise global human water consumption and footprint. Science 350, 1248–1251. doi:10.1126/science.aad1010
-Jaramillo, F., Destouni, G., 2014. Developing water change spectra and distinguishing change drivers worldwide. Geophys. Res. Lett. 41, 8377–8386. doi:10.1002/2014GL061848
-Jaramillo, F., Prieto, C., Lyon, S.W., Destouni, G., 2013. Multimethod assessment of evapotranspiration shifts due to non-irrigated agricultural development in Sweden. J. Hydrol. 484, 55–62. doi:10.1016/j.jhydrol.2013.01.010
-van der Velde, Y., Lyon, S.W., Destouni, G., 2013. Data-driven regionalization of river discharges and emergent land cover–evapotranspiration relationships across Sweden. J. Geophys. Res. Atmospheres 118, 2576–2587. doi:10.1002/jgrd.50224

[revised manuscript text omitted]

time scale during 2000-2006. Based on the hypothesis that the soil surface closely

resembles an uniform height, actively growing grass, completely shading the ground,

potential daily evapotranspiration (PET) was calculated by the FAO Penman–Monteith

205  equation as follows (Allen et al., 1998):

$$PET = \frac{0.408\Delta(R_n - G) + \gamma \frac{900}{T + 273} u_2(e_s - e_a)}{\Delta + \gamma(1 + 0.34u_2)}$$

$$PET = \frac{0.408\Delta(R_n - G) + \gamma \frac{900}{T + 273} u_2(e_s - e_a)}{\Delta + \gamma(1 + 0.34u_2)} \tag{1}$$

where $R_n$ is net radiation at the cover surface (MJ m$^{-2}$ d$^{-1}$), $G$ is soil heat flux (MJ m$^{-2}$ d$^{-1}$), $T$ is mean air temperature (°C), $u_2$ is wind speed (m s$^{-1}$), $e_s$ is saturation vapour

210  pressure (kP$_a$), $e_a$ is actual vapour pressure (kP$_a$), $e_s$–$e_a$ is the saturation vapour pressure

deficit (kP$_a$), $\Delta$ is slope of saturation vapour pressure curve (kP$_a$ °C$^{-1}$), and $\gamma$ is the

psychrometric constant (kP$_a$ °C$^{-1}$). Most sites are in the North Hemisphere except three

EBF sites.

The monthly crop coefficient ($Kc$) is defined as the ratio of the measured total

215  monthly AET and the total monthly PET calculated by Equation 1 varies by month and

vegetation types (Equation 2). The average annual $Kc$ values were calculated using

meanby averaging monthly $Kc$ from January to December for the special siteseach site.

$$Kc = \frac{ET}{ET_0} \quad Kc = \frac{AET}{PET} \tag{2}$$

The LAI time series data for each tower site were downloaded from the Oak Ridge National Laboratory Distributed Active Archive Center (http://daac.ornl.gov/cgi-bin/MODIS/GR_col5_1/mod_viz.html). MODIS LAI data were derived from the fraction of absorbed photosynthetically active radiation (FPAR) that a plant canopy absorbs for photosynthesis and growth in the 0.4–0.7 nm spectral range. The MODIS LAI/FPAR algorithm exploits the spectral information of MODIS surface reflectance at up to seven spectral bands. We extracted monthly LAI data for the periods from 2000 through 2006 across 111 sites using 8-day GeoTIFF data from the Moderate Resolution Imaging Spectroradiometer (MODIS) land subsets' 1-km LAI global fields. We estimated monthly LAI for each flux tower by computing the mean of the 8-day daily values for each month (Fang et al., 2015).

**3. Results**

**3.1. Seasonal variations and long--term means of Kc by land cover**

The average monthly *Kc* based on eddy flux data from 2000 to 2007 increased gradually from January to July and then decreased (Fig. 2). Evergreen broad leaf forest (EBF) had the highest mean monthly *Kc* (97±0.19) (Mean ± standard error) in December (June for sites in the South Hemisphere). *Kc* for both EBF and ENF varied less seasonally than other forest types (Fig. 2). Standard errors for grassland (GRA), evergreen needle leaf forest (ENF) and open shrubland (OS) (0.10-0.17) were larger than other land cover types (0.03-0.10) for April to August. EBF had higher *Kc* for all seasons than other land covers with a peak value of 0.91 (± 0.08) in the winter season (Fig. 3). In winter seasons, cropland (CRO) and OS had the lowest *Kc*, 0.25 (± 0.006) and 0.22 (± 0.004), respectively.

The mean annual *Kc* was 0.39 ($\pm$ 0.04), 0.47 ($\pm$ 0.05), 0.75 ($\pm$ 0.03), 0.45 ($\pm$ 0.02), 0.57 ($\pm$ 0.06), 0.45 ($\pm$ 0.05), and 0.40 ($\pm$ 0.04) for CRO, deciduous broad leaf forest (DBF), EBF, ENF, GRA, mixed forest (MF), and OS, respectively. Yearly average precipitation was higher in EBF and DBF than other land covers (Fig. 4). The precipitation ranking by land cover type was EBF> DBF>  MF> GRA> ENF> CRO> OS. Consequently, OS, MF, GRA, CRO and ENF had relatively lower yearly AET (376-425 mm) than EBF and DBF. Moreover, DBF, EBF and CRO had higher PET than other vegetation surfaces. The variations for monthly AET and PET were presented in Fig. 4 to the contrasting patterns of these two variables. The AET and PET reached maximum value 2.2-3.3 mm d$^{-1}$ and 3.6-4.7 mm d$^{-1}$ at June or July (December or January for the Southern Hemisphere), respectively.

*3.2. Environmental controls on Kc*

As indicated in Equation 1, factors such as temperature and solar radiation were used for PET calculations, and were not independent to AET/PET. Therefore, Since sitedeterminein the same land cover types, so we explored the relationship between *Kc* and site latitude~~.

The results showed that annual *Kc* was negatively ($p<0.05$) correlated with  latitude  (Fig.5) for CRO, DBF, ENF, GRA and MF with a determination coefficient ($R^2$) of 0.83, 0.59 and 0.21, 0.72 and 0.52, respectively. For OS, annual mean

*Kc* also decreased with the increase in site latitude. Most of the study site  fell between 30$^{\circ}$N to 60$^{\circ}$N in latitude.

At the seasonal scale, the linear relationships between monthly *Kc* and total monthly precipitation differed among different land cover types (Fig. 6). Monthly *Kc* increased with monthly precipitation in the same ecosystem type with the $R^2$ ranking from high to low: OS>MF>GRA>ENF>CRO>DBF. The monthly *Kc* for open shrublands (OS) was especially sensitive to precipitation ($R^2$= 0.69, *p*<0.001). The monthly *Kc* for EBF was not as sensitive to precipitation as other ecosystems because EBF was generally found in a wet environment with a peak monthly precipitation of 468 mm. Moreover, *Kc* for OS, GRA and MF in relatively drier environments had lower values (Fig. 2). Therefore, *Kc* was closely related to the monthly precipitation.

In addition to growing season, site latitude and monthly precipitation, leaf area index affected the monthly *Kc* (Fig. 7). *Kc* was obviously influenced by  leaf area index (LAI) for all land covers except EBF. The determination coefficients for different land covers were OS> MF=GRA> ENF>DBF>CRO. The LAI range was up to 6 m$^2$ m$^{-2}$ in most land covers, while it only reached 3-4 m$^2$ m$^{-2}$ in OS and CRO.

*3.3. Kc models*

A series empirical *Kc* models have been developed using a multiple linear regression approach with precipitation, leaf area index (LAI), and site latitude as independent variables (Table 1). The monthly precipitation, LAI and site latitude influence *Kc* (*p*<0.1) for most ecosystems studied in different seasons except at EBF in summer and fall, and for OS in the spring. As annual

precipitation increases, total leaf area increases, therefore *Kc* increases for ENF in all seasons and most of the time for DBF and MF. As site latitude increases, *Kc* values are found to decrease in some periods at CRO, DBF and MF sites. In addition, *Kc* is closely correlated to LAI, site latitude, and monthly precipitation at ENF in fall and OS in winter with $R^2$ 0.55 and 0.99. All land covers have peak values (0.53 ± 0.04-1.01 ± 0.17) in the summer months. Except for EBF and GRA, *Kc* values have a close relationship with the monthly precipitation in the summer with $R^2$ ranging from 0.21 to 0.90. The linear relationships are significant for most vegetation types, suggesting that the regression models (Table 1) can be used to estimate monthly *Kc* if LAI and precipitation for a specific ecosystem are available.

*3.4. The validation of the regression models of Kc*

All *Kc* multiple regression models for different seasons were validated by ecosystem type (Fig. 8). The model validation was carried out for 30 sites at a monthly scale. The results showed that the modeled AET calculated from the multiple *Kc* models compared well to measurements with  $R^2$ ranging 0.28-0.56. Among the ecosystems, the model for DBF appeared to be the most accurate one with a $R^2$ of 0.56. However, model validation results for CRO, EBF and OS were not as satisfactory as indicated by the slopes (<1.0 or >1.0) of the regression equations.

**4. Discussion**

Our study estimated annual and seasonal crop coefficient (*Kc*) for seven land cover types using measured global eddy flux data. We comprehensively evaluated environmental controls (i.e., precipitation, LAI, and site latitude) on annual and growing season

*Kc* and developed a series of multiple linear regression models that can be used for estimating monthly AET over time and space for some vegetation types.

*4.1. Crop coefficient variation in different seasons*

Several recent studies had shown that *Kc* reached the maximum value in the middle of the growing season in many ecosystems, such as a *P. euphratica* forest in the riparian area (Hou et al., 2010)(Hou et al., 2010) in a desert environment, a watermelon crop covered with plastic mulch in Florida (Shukla et al., 2014b;Shukla et al., 2014a)(Shukla et al., 2014a; Shukla et al., 2014b), soybean in Nebraska (Irmak et al., 2013b)(Irmak et al., 2013b), a temperate desert steppe in Inner Mongolia(Zhang et al., 2012)(Zhang et al., 2012). As Fig. 2 shows, most of the land covers hadhave peak *Kc* during June to August, (In the Northern Hemisphere), while the seasonal patterns of ENF and EBF variedvary less than other surfaces. Vegetation growth for both the ENF and EBF sites is active throughout the year. The mean crop coefficientscoefficient for early period-mid-density fruit trees in the early growing season is about 0.5 (Allen et al., 1998;; Allen and Pereira, 2009) which is similar to those found for DBF or MF during April and May. In addition, the middle season *Kc* values for apple and peach trees with active ground cover were higher than *Kc* for DBF sites during the summer. It is likely that the orchards had higher evapotranspiration rates than natural forests due to irrigation in orchards. We also find that the CRO has relatively low precipitation with a high PET because of irrigation. The irrigation has been proven to be a determine factor to AET at the local and even at the global scale (Jaramillo and Destouni, 2015). Thus, the *Kc* for CRO mainly depends on the irrigation schedule and the primary crops. The loss of leaves on DBF and MF lead to an obvious larger stand error for *Kc* in fall (Fig. 3). The soil water evaporation represents the

main water loss, thus key component of *Kc* when the ecosystems lack of leaves or plants in winter (Allen et al., 1998). Moreover, the AET/PET is biologically meaningful in vegetation type distribution (Stephenson, 1998), thus, when LAI becomes small for DBF during winter, the AET/PET reflects the characteristics of evaporation capacity for the ground surface.

*4.2. Environmental control factors for Kc*

The ecosystem covers and the distributions of the vegetation classes wereare determined by the latitude (Potter et al., 1993).(Potter et al., 1993). Crop coefficient varies predominately by ecosystems, *Kc* will in most cases increaseand *Kc* increases as the site latitude decreaseddecreases for the same land cover type (Fig. 5). As the latitude decreaseddecreases, the increasing temperature and the solar radiation increased andresults of PET increasing, thus, the acceleration for AET should be faster than PET. The reason may be the vegetation characteristics would beare different for the same land cover type. in different latitudes. Models developeddevelop from the FLUXNET data may be best used on flat areas for a givenspecific latitude given that eddy covariance towers were generally installed on flat lands (Baldocchi et al., 2001)(Baldocchi et al., 2001). For areas with complex topography, the relationship between *Kc* and site latitude may be more complicated.

Spatial variations of *Kc* are characteristic of ecosystems, but *Kc* is also affectedeffected by climate factors such as rainfall. For example, *Kc* was highly correlated with precipitation for most land covers (Fig. 6).The rainfall is the major source of soil water and AET in natural ecosystems (Parent and Anctil, 2012)(Parent and Anctil,

2012). During dry years or periods, a lack of precipitation may cause a reduction of the leaf area index and $Kc$ will decrease to response the ecosystem function.. During rainy seasons, as, leaf area index and stomatal conductance of trees and rain-fed crops increases, so does $Kc$ (Kar et al., 2006;Zeppel et al., 2008)(Kar et al., 2006; Zeppel et al., 2008). Irrigation of cropland is a primary mechanism for increasing yield (Du et al., 2015;Fereres and Soriano, 2007)(Fereres and Soriano, 2007; Du et al., 2015), so the CRO may have a high monthly $Kc$ even at sites with a low precipitation. In contrast, $Kc$ does not have a close relationship with precipitation under a wet environment. For example, the EBF site had a monthly precipitation as high as 468 mm/month and generally exceeded monthly AET. In an opposite case for the OS sites, monthly precipitation values were between 0.7 to 69 mm, and $Kc$ was highly correlated to monthly precipitation. Moreover, the soil moisture could be a limiting factor to AET, and would affect $Kc$ in dry periods. When the time lag between precipitation and soil moisture might cause errors in calculating AET and modeling $Kc$ in the long dry or wet season. However, at the monthly scale, previous modeling work (Fang et al., 2015) suggestsuggests that considering a time lag does not increase the prediction power dramatically (G. Sun Personal communication).

Besides precipitation, leaf area index (LAI) also affects $Kc$ in dry and semi-humid areaareas (Zhang et al., 2012;Kang et al., 2003)(Kang et al., 2003; Zhang et al., 2012). Unlike precipitation, LAI directly affects $Kc$ in AET calculations (Novák, 2012;Tolk and Howell, 2001). Inter annual $Kc$ values are stable at the GRA and OS sites due to the steady seasonal LAI between years while the plantation forest sites had a more dynamic LAI pattern(Marsal et al., 2014a). As the growth rate of the perennial plants could have

375

. Unlike precipitation, LAI directly affects *Kc* in AET calculations (Tolk and Howell, 2001; Novák, 2012). Inter-annual *Kc* values are stable at the GRA and OS sites due to the steady seasonal LAI between years while the plantation forest sites had a more dynamic

380 LAI pattern(Marsal et al., 2014a). As the growth rate of the perennial plants could have large effects on the relationship between *Kc* and LAI, long-term data are needed to estimate *Kc* as a function of all environmental factors.

*4.3. Modeling the dynamics of Kc*

Our study results are consistent with previous studies that show that the growing stage is

385 a key factor for estimating *Kc* in agricultural crops (Allen et al., 1998; Zhang et al., 2013; Alberto et al., 2014; Wei et al., 2015), fruit trees (Abrisqueta et al., 2013; Marsal et al., 2014b), salt grass (Bawazir et al., 2014) and *Populus euphratica Oliv* forest (Hou et al., 2010). Additionally, our study showed that *Kc* fluctuated more

390 dramatically in DBF, GRA, and MF than other land covers in different seasons (Table 1). Studies also show that monthly leaf stomatal resistance that varies over time is important in estimating the seasonal crop coefficient for a citrus orchard (Taylor et al., 2015). The LAI and total monthly precipitation were considered as independent factors (Bond-Lamberty and Thomson,

395 2010) and both of them varied in both time and space while the site latitude only represents spatial influences on *Kc*. The modeled AET was acceptable for

land cover typesDBF, ENF, GRA and MF (Fig. 8), and could be used for monthly AET calculation for large spatial scale and homogeneous ecosystems. The slope of CRO modeling ET to AET was 50 percent lower from 1:1 line may be because the crops was irrigated when the soil lack of water content. Meanwhile, the OS has a large proportion of bare soil with low soil water content may result of an overestimate in modeling ET. The lack of sites samples may cause a low accuracy of validation in OS and EBF molding ET. Thus, the multiple linear regression equations developed from this study take account of both spatial and temporal changes in land surface characteristics and offer a powerful tool to estimate of seasonal dynamicdynamics of $Kc$ for differentmost ecosystems (Table 1).

**5. Conclusions**

To seek a convenient method to calculate monthly AET inat large spatial scalescales, we comprehensively examined the relations between $Kc$ and environmental factors using eddy flux data from 81 sites (mainly in the northern hemisphere) with different land covers. We found that $Kc$ values varied largely among CRO, DBF, EBF, GRA and MF and over seasons. PrecipitationBesides EBF, precipitation determined $Kc$ in the growing seasons (such as summer), and was chosen as a key variable to calculate $Kc$. We established multiple linear equations for different land covers and seasons to model the dynamics of $Kc$ as function of LAI, site latitude and monthly precipitation. These empirical models could be helpful in calculating monthly AET at the regional scalesscale with readily available climatic data and vegetation structure information. Our study extended the applications of the traditional $Kc$ method for estimating crop water use to estimating AET rates and evaporative stress for natural ecosystems. Future studies should

further test the applicability of the empirical *Kc* models under extreme climatic

conditions and for those under-represented ecosystems by the FLUXNET.

**Acknowledgements**

We are grateful for grants from the National Natural Science Foundation of China (51309132), for supporting this collaborative work between Nanjing University of Information Science and Technology and the Eastern Forest Environmental Threat Assessment Center at the USDA Forest Service Southern Research Station. This work used eddy covariance data acquired by the FLUXNET community and in particular by the following networks: AmeriFlux [U.S. Department of Energy, Biological and Environmental Research, Terrestrial Carbon Program (DEFG02-04ER63917 and DE-FG02-04ER63911)], AfriFlux, AsiaFlux, CarboAfrica, CarboEuropeIP, CarboItaly,

CarboMont, ChinaFlux, Fluxnet-Canada (supported by CFCAS, NSERC, BIOCAP, Environment Canada, and NRCan), GreenGrass, KoFlux, LBA, NECC, OzFlux, TCOS-Siberia, and the United States China Carbon Consortium (USCCC). We acknowledge the financial support to the eddy covariance data harmonization provided by CarboEuropeIP, FAO-GTOS-TCO, iLEAPS, Max Planck Institute for Biogeochemistry, National Science Foundation, University of Tuscia, Université Laval and Environment Canada, and U.S. Department of Energy, and the database development and technical support from Berkeley Water Center, Lawrence Berkeley National Laboratory, Microsoft Research eScience, Oak Ridge National Laboratory, University of California, Berkeley, and University of Virginia. This work also used MODIS land subset (Oak Ridge National Laboratory Distributed Active Archive Center (ORNL DAAC). 2011. MODIS subsetted

land products, Collection 5). We also thank the reviewers and associate editor for their constructive comments on the manuscript.

**References**

[revised manuscript text omitted]

$p{<}0.001$, $p{<}0.01$, $p{<}0.1$., and the blank spaces mean non-significant. In the North Hemisphere, Spring is the month of February, March and April; Summer is the month of May, June and July; Fall is August, September and October; Winter is November, December and January. In the South Hemisphere, Spring is August, September and October; Summer is November, December and January; Fall is February, March and April; and winter is May, June and July.

**Figure captions**

Fig. 1 Location of eddy flux sites from which climate and evapotranspiration data are collected.

Fig. 2 The variation of *Kc* for the different IGBP codes. The error bars are standard errors among different sites. The seven vegetation covers are Open shrubland (OS), Cropland (CRO), Grassland (GRA), Deciduous broad leaf forest (DBF), Evergreen needle leaf forest (ENF) and Evergreen broad leaf forest (EBF), and Mixed forest (MF). For sites in the South Hemisphere, July data were plotted as in January.

Fig.3 Average *Kc* at spring, summer, fall and winter in different vegetation types. The error bars are standard errors among different sites. Spring is the month of February, March and April; Summer is the month of May, June and July; Fall is August, September and October; Winter is November, December and January. In the South Hemisphere, Spring is August, September and October; Summer is November, December and January; Fall is February, March and April; and winter is May, June and July.

Fig. 4  Monthly AET and PET, and annual total precipitation (P), AET and PET for different vegetation types. The error bars are standard errors among different sites.

Fig. 5 Variation of annual *Kc* at different latitude (Lat). (a)  cropland (CRO), deciduous broad leaf forest (DBF), evergreen broad leaf forest (EBF), and (b) evergreen needle leaf forest (ENF), grassland (GRA), mixed forest (MF), and open shrubland (OS). The absolute values of the latitude were used in EBF for sites in the Southern Hemisphere and all the determination coefficients ($R^2$) listed in the figure were significant ($p<0.05$).

Fig. 6 Relationships between the average monthly *Kc* and  monthly precipitation (P, mm) for different vegetation surfaces. Figures (a)~(g) represent cropland (CRO), deciduous broad leaf forest (DBF), evergreen broad leaf forest (EBF), evergreen needle leaf forest (ENF),

grassland (GRA), mixed forest (MF), and open shrubland (OS), respectively. All the determination coefficients ($R^2$) listed in the figure were significant ($p<0.001$).

665 Fig. 7 Relationships between the average monthly $Kc$ and leaf area index for different vegetation surfaces. Figures (a)~(g) stand for cropland (CRO), deciduous broad leaf forest (DBF), evergreen broad leaf forest (EBF), evergreen needle leaf forest (ENF), grassland (GRA), mixed forest (MF), and open shrubland (OS). All the determination coefficients ($R^2$) listed in the figure were significant ($p<0.$05).

670 Fig. 8 Relationships between the simulated ET using $Kc$ from Table 1 (SET) and the measured ET (AET) for different vegetation surfaces. Figures (a)~(f) stand for cropland (CRO), deciduous broad leaf forest (DBF), evergreen broad leaf forest (EBF), evergreen needle leaf forest (ENF), grassland (GRA), mixed forest (MF), and open shrubland(OS). All the determination coefficients ($R^2$) listed in the figure were significant ($p<0.001$).

675

[Figure]

680    Fig. 1 Location of eddy flux sites from which climate and evapotranspiration data are collected.

[Figure]

Fig. 2 The variation of *Kc* for the different IGBP codes. The error bars are standard

685    errors among different sites. The seven vegetation covers are Open shrubland (OS), Cropland

(CRO), Grassland (GRA), Deciduous broad leaf forest (DBF), Evergreen needle leaf forest (ENF)

and Evergreen broad leaf forest (EBF), and Mixed forest (MF). For sites in the South Hemisphere,

July data were plotted as in January.

[Figure]

690

Fig.3 Average *Kc* at spring, summer, fall and winter in different vegetation types. The error bars are standard errors among different sites. Spring is the month of February, March and April; Summer is the month of May, June and July; Fall is August, September and October;

Winter is November, December and January.

695   In the South Hemisphere, Spring is August, September and October; Summer is November, December and January; Fall is February, March and April; and winter is May, June and July.

[Figure]

[Figure]

[Figure]

[Figure]

700

Fig.4  Monthly AET and PET, and annual total precipitation (P), AET and PET for different vegetation types. The error bars are standard errors among different sites.

[Figure]

 Fig. 5 Variation of annual $Kc$ at different latitude (Lat). (a)  cropland (CRO), deciduous broad leaf forest (DBF), evergreen broad leaf forest (EBF), and (b) evergreen needle leaf forest (ENF), grassland (GRA), mixed forest (MF), and open shrubland (OS). The absolute values of the latitude were used in EBF for sites in the  Southern Hemisphere and all the determination coefficients ($R^2$) listed in the figure were significant ($p<0.05$).

710

[Figure]

[Figure]

 Fig. 6 Relationships between the average monthly *Kc* and  monthly precipitation (P, mm)

for different vegetation surfaces. Figures (a)~(g) represent  cropland (CRO), deciduous broad

leaf forest (DBF), evergreen broad leaf forest (EBF), evergreen needle leaf forest (ENF),

grassland (GRA), mixed forest (MF), and open shrubland (OS), respectively. All the

determination coefficients ($R^2$) listed in the figure were significant (*p*<0.001).

[Figure]

725

[Figure]

Fig. 7 Relationships between the average monthly $Kc$ and leaf area index for different vegetation surfaces. Figures (a)~(g) stand for cropland (CRO), deciduous broad leaf forest (DBF), evergreen broad leaf forest (EBF), evergreen needle leaf forest (ENF), grassland (GRA), mixed forest (MF), and open shrubland (OS). All the determination coefficientcoefficients ($R^2$) listed in the figure were significant ($p<0.05$)).

[Figure]

[Figure]

Fig. 8 Relationships between the simulated ET using *Kc* from Table 1 (SET) and the measured
ET (AET) for different vegetation surfaces. Figures (a)~(f) stand for cropland (CRO), deciduous
broad leaf forest (DBF), evergreen broad leaf forest (EBF), evergreen needle leaf forest (ENF),
grassland (GRA), mixed forest (MF), and open shrubland(OS). All the determination
coefficients ($R^2$) listed in the figure were significant ($p<0.001$).